# Computational Analysis of the Tripartite Interaction of Phasins (PhaP4 and 5)-Sigma Factor (σ^24^)-DNA of *Azospirillum brasilense* Sp7

**DOI:** 10.3390/polym16050611

**Published:** 2024-02-23

**Authors:** Yovani Aguilar-Carrillo, Lucía Soto-Urzúa, María De Los Ángeles Martínez-Martínez, Mirian Becerril-Ramírez, Luis Javier Martínez-Morales

**Affiliations:** Centro de Investigaciones en Ciencias Microbiológicas, Instituto de Ciencias, Benemérita Universidad Autónoma de Puebla, Av. San Claudio y Av. 24 Sur, Col. San Manuel Ciudad Universitaria, Puebla 72570, Mexico; yovani.aguilarc@alumno.buap.mx (Y.A.-C.); lucia.soto@correo.buap.mx (L.S.-U.); angeles.martinezm@correo.buap.mx (M.D.L.Á.M.-M.); mirian.becerrilr@alumno.buap.mx (M.B.-R.)

**Keywords:** *Azospirillum brasilense* Sp7, stress conditions, phasins, σ^24^ factor, molecular docking

## Abstract

*Azospirillum brasilense* Sp7 produces PHB, which is covered by granule-associated proteins (GAPs). Phasins are the main GAPs. Previous studies have shown phasins can regulate PHB synthesis. When *A. brasilense* grows under stress conditions, it uses sigma factors to transcribe genes for survival. One of these factors is the σ^24^ factor. This study determined the possible interaction between phasins and the σ^24^ factor or phasin-σ^24^ factor complex and DNA. Three-dimensional structures of phasins and σ^24^ factor structures were predicted using the I-TASSER and SWISS-Model servers, respectively. Subsequently, a molecular docking between phasins and the σ^24^ factor was performed using the ClusPro 2.0 server, followed by molecular docking between protein complexes and DNA using the HDOCK server. Evaluation of the types of ligand–receptor interactions was performed using the BIOVIA Discovery Visualizer for three-dimensional diagrams, as well as the LigPlot server to obtain bi-dimensional diagrams. The results showed the phasins (Pha4_Abs7_ or Pha5_Abs7_)-σ^24^ factor complex was bound near the −35 box of the promoter region of the *pha*C gene. However, in the individual interaction of PhaP5_Abs7_ and the σ^24^ factor, with DNA, both proteins were bound to the −35 box. This did not occur with PhaP4_Abs7_, which was bound to the −10 box. This change could affect the transcription level of the *pha*C gene and possibly affect PHB synthesis.

## 1. Introduction

The α-proteobacteria *Azospirillum brasilense* Sp7 is considered as a Plant Growth Promoting Rhizobacteria (PGPR). The PGPR activity occurs through nitrogen fixation and the production of bacterial phytohormones. *A. brasilense* is an epiphytic strain capable of colonizing the host plant alone [1].

*A. brasilense* Sp7 is a Gram-negative bacterium characterized by producing poly-3-hydroxybutyrate (PHB) [2]. PHB is a biopolymer that is part of a family of microbial thermoplastic polyesters [2,3]. PHB is used as an energy and carbon storage compound when bacteria are exposed to nutrient-limiting conditions (N, O, and P) and environmental stresses. The main factor involved in controlling PHB synthesis is the C:N ratio of the growth medium [4,5]. Due to the increase in the use of plastics around the world, PHB has become more relevant. The above is due to the properties of PHB, which are similar to the physicochemical characteristics of petroleum-derived plastics. Additionally, PHB is a biocompatible and biodegradable polymer, even in marine environments [4,5,6,7,8].

The biosynthesis of PHB depends on the activity of three enzymes, β-ketothiolase (PhaA), acetoacetyl-CoA reductase (PhaB), and PHA synthase (PhaC), which are encoded by the *pha*A, *pha*B, and *pha*C genes, respectively [8]. In the first step of PHB biosynthesis, PhaA condenses two acetyl-CoA molecules into acetoacetyl-CoA. Subsequently, PhaB reduces acetoacetyl-CoA to [R]-3-hydroxybutyryl-CoA. And finally, PhaC polymerizes [R]-3-hydroxybutyryl-CoA to PHB (Figure 1) [2].

As PHB biosynthesis begins, biopolymer accumulates in the cytoplasm as spherical water-insoluble inclusions (called carbonosomes), which are variable in size and number according to the producer microorganism as well as the conditions of bacterial growth [6,9]. The carbonosome membrane contains 0.5% phospholipids, 2% granule-associated proteins (GAPs), and approximately 98% (*w*/*w*) PHB [5]. Currently, the presence of four types of GAP has been determined: PHA synthases (PhaC), PHA depolymerases (PhaZ), regulators (PhaR or PhaM), and phasins (PhaP) [6,9,10]. The last ones are the most abundant polypeptides in the carbonosome membrane and are involved in coating and stabilizing PHB chains due to their small size and amphiphilic properties [11]. Initially, phasins were involved in regulating the number and size of carbonosomes. However, recent studies suggest activities such as mediating the binding of PHB to the nucleoid region and promoting bacterial resistance under stress conditions [12]. In *A. brasilense* Sp7, six phasins, PhaP1_Abs7_ to PhaP6_Abs7_, have been reported [4]. The predicted structures for these proteins showed a high percentage of amino acids in α-helix conformation and disordered regions, which provide flexibility to the protein [4,13]. In addition, phasins have leucine zipper motifs, which are involved in the formation of oligomers [6].

Production and storage of PHB provides enough energy to continue with the metabolic processes, enabling bacterial survival under stress conditions [14,15]. Bacterial growth is affected by several conditions (e.g., ultraviolet radiation, heat, osmotic shock, desiccation, and oxidative stresses, among others) that affect gene expression [1,7]. *A. brasilense* uses some transcription factors to change its genetic expression pattern in response to stressful conditions [16]. In *A. brasilense* Sp7, some transcription factors have been reported, for example, σ^70^ (RpoD), σ^54^ (RpoN), σ^32^ (RpoH), σ^24^ (RpoE), and the FeCl sigma factor [1,16,17]. Genetic expression mediated by sigma factors is possible because it allows binding of the RNAP sequence (RNA Polymerase Recognition) to the promoter DNA regions [18]. Alternative sigma factors, such as the σ^24^ factor, allow *A. brasilense* Sp7 to repair the cellular damage caused by oxidative and photo-oxidative stresses [1,16,17,19].

Under stressed growth conditions, *A. brasilense* activates gene transcription for PHB biosynthesis, which is essential. The *pha*C and *pha*Z genes increase their transcription levels in those conditions, suggesting the importance of PHB synthesis as a protective mechanism for bacterial survival [14].

Bacteria can detect fluctuations in their environment. These varieties can activate or inactivate system components that switch on or switch off the RNA polymerase activity to regulate the transcription of genes related to the cellular response [17]. For this reason, studies on the ability of phasins to act as regulators can be essential to understanding transcription genes. This might depend on whether phasins act as individual proteins that detect signals and can affect the cellular response based on their structure, since they have a sensory and a regulatory domain on the same protein [17].

The understanding of how transcription factors bind to the promoter regions and promote or silence the DNA transcription is increasing. In this study, we evaluate the possible interactions occurring around DNA transcription. DNA-binding regions are different in every living organism. Despite the above, there are well-characterized motifs for the binding of the σ^70^, σ^54^, σ^28^, and σ^32^ factors to the −10 and −35 regions in *Escherichia coli*. However, due to the low conservation of the σ^24^ and σ^28^ factors, it has not been possible to find a well-characterized motif for DNA binding [18,20]. Also, it is necessary to consider the DNA sequences for interaction with the sigma factors [17]. In consequence, the prediction of these motifs is complicated. Nevertheless, software and servers, such as the MEME Suite server, could predict the motifs for the interaction between DNA and transcription factors [21].

The goal of this computational study was to analyze the possible interactions between the phasins PhaP4_Abs7_ and PhaP5_Abs7_ with the promoter region of the *pha*C gene and with the σ^24^ factor (RpoE).

## 2. Materials and Methods

### 2.1. Three-Dimensional Analysis of the Structures of PhaP4_Abs7_, PhaP5_Abs7_, and σ^24^

The three-dimensional structures of PhaP4_Abs7_ and PhaP5_Abs7_ were predicted by the Iterative Threading ASSEmbly Refinement (I-TASSER) server [22,23,24,25]. The crystal structure of *Aeromonas hydrophila* PhaP_Ahy_ chain A (PDB number 5IP0) was used as a template [26]. For the σ^24^ factor, the three-dimensional structure was generated using the SWISS-MODEL server [27]. The crystal structure of *Rhodobacter sphaeroides* SigE, chain A (PDB number 2Q1Z), was used as a template [28]. As a second step, using the Alpha Fold2 server [29], the three-dimensional structures of the three proteins were generated to compare models between the servers and choose the models with the best scores to continue with the molecular docking. To evaluate the quality of the three-dimensional structures, the SWISS-MODEL structure evaluation server was used, and the Qualitative Model Energy ANalysis (QMEAN) and its Ramachandran plots were obtained [30].

### 2.2. Determination of the Promoter Regions (−10 and −35) Upstream of the phaC Gene

The promoter region of the *pha*C was determined using the MEME Suite server [21]. The 236-bp upstream region of the *pha*C gene (locus tag: AMK58_RS06750) was selected, which included the start codon (ATG) of the *pha*C gene. The upstream region comprised an intergenic region of divergent genes.

### 2.3. Molecular Docking to Form the Phasin-σ^24^ Factor Protein Complex and Individual Docking of Phasins and the σ^24^ Factor with DNA

The formation of the protein complex between phasins and the σ^24^ factor using the ClusPro 2.0 server [31,32,33,34] was determined, considering the phasins as ligands and the σ^24^ factor as a receptor. The HDOCK server was used [35,36,37,38], which led us to evaluate the interaction between phasins and DNA (upstream region of the *pha*C gene). It is necessary to mention that every single phasin was considered a ligand, and the σ^24^ factor was considered a receptor. Finally, the HDOCK server was also used to analyze the interaction between the σ^24^ factor and DNA. The first was probed as a ligand, and DNA as a receptor.

### 2.4. Tripartite Molecular Docking between PhaP4_Abs7_-σ^24^ and PhaP5_Abs7_-σ^24^ Protein Complexes with the Promoter Region of the phaC Gene

Using the HDOCK server, the molecular docking was carried out between the protein complexes formed by the interaction between the σ^24^ factor and phasin (σ^24^-phasin) with the DNA region (upstream region of the *pha*C gene). DNA was considered as a receptor, and the protein complex was used as the ligand.

Three coupled interactions, phasin-σ^24^ factor, phasin-DNA or σ^24^ factor-DNA, and phasin-σ^24^-DNA, were visualized using the BIOVIA Discovery Studio Visualizer. In all interactions, the start codon (ATG) of the *pha*C gene was used as a visual reference point in the software [39]. Bi-dimensional diagrams of these three interactions were obtained using the LigPlot server [40].

## 3. Results

There is insufficient information about the phasins in *A. brasilense* Sp7. Therefore, in this study, three-dimensional models of the *A. brasilense* phasins were obtained using the methodology reported by Martínez-Martínez et al., 2019 [4]. The crystal structure of *A. hydrophila* PhaP_Ahy_ was used as a template to obtain the three-dimensional structures of the *A. brasilense* Sp7 phasins on the I-TASSER server (Figure 2A,B). The SWISS-MODEL server was used to obtain the three-dimensional structure of the σ^24^ factor, and the *R. sphaeroides* SigE served as a template (Figure 2C). At the same time, we utilized Alpha Fold2 to obtain three-dimensional models of phasins and the σ^24^ factor, compared them with the other models, and used the best models with the most acceptable QMEAN values between servers.

The QMEAN results were acceptable for PhaP5_Abs7_ and the σ^24^ factor, with a value of −2.25 and −1.84, respectively. However, for PhaP4_Abs7_, the quality was not as good as was expected, but acceptable (Figure 2A–C). Nevertheless, the three-dimensional models performed with Alpha Fold2 obtained better QMEAN scores in comparison with models from I-TASSER and SWISS-MODEL, with values of −0.07 for PhaP4_Abs7_, −1.64 for PhaP5_Abs7_, and −1.57 for the σ^24^ factor (Figure 2D–F). For this reason, for protein–protein and protein–DNA, molecular docking was performed using three-dimensional models obtained by the Alpha Fold2 server. Finally, a structural superimposition was performed, suggesting that the two three-dimensional models of each protein obtained by the servers might be structurally very similar (Appendix A).

Simultaneously, a multiple alignment was carried out in Clustal omega between the *A. brasilense* Sp7 phasins and the *A. hydrophila* phasin (PDB number 5IP0). The results showed that *A. brasilense* phasins have a percentage identity correlation of 11.90% for PhaP4_Abs7_ and 25.49% for PhaP5_Abs7_ to the *A. hydrophila* phasin. PhaP4_Abs7_, PhaP5_Abs7,_ and PhaP_Ahy_ are 188, 171, and 116 amino acids in length, respectively. In addition, the *A. brasilense* σ^24^ factor showed 43.48% identity correlation to *R. sphaeroides *SigE (Appendix A).

The promoter region of the *pha*C gene was determined using the MEME Suite server. The promoter region was selected considering the upstream region of the *pha*C gene. It corresponds to 233 bp, plus the start codon (ATG) of the *pha*C gene as a reference point. The bp number chosen corresponds to an intergenic region of the divergent genes, between the *pha*C and the AMK58_RS06755 genes. The MEME Suite server reads DNA from 5′ to 3′. Therefore, the −10 and −35 boxes were selected in agreement with the closest sequence to these positions. The motif for the −10 box, containing the sequence ACCGCAC, was located at position 225 to 231 bp, while the −35 box, with the sequence CCTAAA, was located at position 200 to 205 bp. The motif with the sequence TGTATRA was not considered in this study as a possible interaction motif, since it was not close to positions −10 or −35 or nearby (Figure 3).

Protein–protein molecular docking was performed using the ClusPro 2.0 server to evaluate the interactions between each of the two phasins (PhaP4_Abs7_ and PhaP5_Abs7_) and the σ^24^ factor (PhaP4_Abs7_-σ^24^ and PhaP5_Abs7_-σ^24^). However, the HDOCK server allowed us to analyze the molecular docking on the individual interactions of every single phasin and the σ^24^ factor with the DNA (upstream region of the *pha*C gene) (PhaP4_Abs7_-DNA, PhaP5_Abs7_-DNA, and σ^24^-DNA). Subsequently, the server was also used for the molecular docking between the protein complexes (PhaP4_Abs7_-σ^24^ and PhaP5_Abs7_-σ^24^) and DNA. Finally, the types of interactions in the three-dimensional models were analyzed by the BIOVIA Discovery Studio Visualizer. In addition, to allow a more detailed analysis, the LigPLot server was used to verify the interactions using two-dimensional schemes.

In the analysis of the interactions in the PhaP4_Abs7_-σ^24^ and PhaP5_Abs7_-σ^24^ protein complexes (Figure 4), the σ^24^ factor was considered as the receptor and the phasins as ligands. The results showed that phasins were bound to the C-terminal and N-terminal sides of the σ^24^ factor. The PhaP4_Abs7_ phasin bounds the N-terminal and C-terminal regions of the σ^24^ factor, simulating a “lid”, because the phasin covers the concavity of the σ^24^ factor. However, PhaP5_Abs7_ was reported to internalize itself in the concavity of the σ^24^ factor. PhaP5_Abs7_ enters on its N-terminal side, forming a plug when it binds with the σ^24^ factor (Figure 4). All analyzed protein complexes (PhaP4_Abs7_-σ^24^ and PhaP5_Abs7_-σ^24^) showed interactions that could be stable, possibly due to the hydrogen bonds formed between the two proteins (Figure 5 and Table 1). In addition, other less common interactions that contribute to stabilizing the protein complex are possible (Appendix A).

Subsequently, interaction analyses were carried out between DNA docked with every single phasin and DNA docked with the σ^24^ factor. To visualize and compare the changes generated in every interaction (DNA-phasin or DNA-σ^24^), the phasin or the σ^24^ factor were considered ligands. Finally, the molecular docking was performed between the PhaP4_Abs7_-σ^24^ and PhaP5_Abs7_-σ^24^ protein complexes with DNA, where the protein complex functioned as a ligand and the DNA was the receptor in both situations. The HDOCK server allowed us to visualize the protein–DNA molecular docking, since this software allows molecular docking between large regions of DNA and proteins. PhaP4_Abs7_ binds to DNA near the −10 box. It showed interactions of PhaP4_Abs7_ with the nucleotides of the 5′ to 3′ strand with G220, A221, A222, G223, G224, A225, and C226, while on the 3′ to 5′ strand, PhaP4_Abs7_ binds with G244, C245, G246, T248, and C249. PhaP5_Abs7_ binds to the promoter region and binds to nucleotides in the −35 box. PhaP5_Abs7_ interacts on the 5′ to 3′ strand with C194, T195, A203, A204, A205, and A206, while on the 3′ to 5′ strand, it binds to A271, G272, G273, C274, T276, and C277. In addition to the individual interactions, the σ^24^ factor binds near the −35 box, interacting on the 5′ to 3′ strand with C201, T202, A203, T208, C209, and G210, and on the 3′ to 5′ strand with G266, T267, T268, T269, and T270 (Figure 6 and Figure 7). The main types of interactions were hydrogen bonds, attractive charges, and salt bridges (Table 2 and Appendix A).

After performing the molecular docking between the protein complex and the DNA, a change was observed near the −35 box in the binding site for the PhaP4_Abs7_-σ^24^ complex. The last binds in the 5′ to 3′ strand with the nucleotides A203, A204, A205, T213, A214, and A215 in the DNA and in the 3′ to 5′ strand with the nucleotides T254, T255, G262, C263, G264, A265, C274, T275, and T276. PhaP4_Abs7_ binds individually near the −10 box. However, the PhaP5_Abs7_-σ^24^ protein complex binds near the −35 box, on the leading strand with the nucleotides G199, C201, T202, and A203, and on the 3′ to 5′ strand with G266, T267, A271, C277, A278 and G279. The PhaP5_Abs7_-σ^24^ protein complex maintained its binding near the −35 box in the upstream region of the *pha*C gene. Similar behavior was shown for each DNA–phasin individual interaction (Figure 8 and Figure 9). In this molecular docking, attractive charges and hydrogen bonds were mainly detected (Table 3 and Appendix A).

## 4. Discussion

Since there are no crystallographic or nuclear magnetic resonance (NMR) structures of the PhaP4_Abs7_, PhaP5_Abs7,_ and σ^24^ factor of *A. brasilense* Sp7, we developed three-dimensional models of these proteins. On the I-TASSER server, the three-dimensional models of phasins were designed, whereas the SWISS-MODEL server was used to obtain the σ^24^ factor structure. The QMEAN results were favorable for PhaP5_Abs7_ and the σ^24^ factor, with values of −2.25 and −1.84, respectively. However, the QMEAN scores for PhaP4_Abs7_ showed low quality, with a score of −7.42. Therefore, we used the AlphaFold2 server to obtain the other models. After using the AlphaFold2 server, we obtained a model with higher quality for PhaP4_Abs7_, with a QMEAN value of −0.07, whereas for PhaP5_Abs7_ and the σ^24^ factor, the QMEAN values also improved, with −1.64 and −1.57, respectively. The data indicate that scores closest to 0.0 points have a “native” structure, and as a rule of thumb, a QMEAN score below −4.0 indicates a low quality of modeling [30].

Likewise, with the I-TASSER and SWISS-MODEL models, the favoring percentages in the Ramachandran graphs were good for the amino acids of the phasin PhaP5_Abs7_, with 98.82%, and for the σ^24^ factor, with 93.14%. However, Ramachandran graphs were not as favorable for PhaP4_Abs7_, which showed 78.41% (Figure 2A–C). Nevertheless, with Alpha Fold models, the percentages in the Ramachandran graphs were good for all models, with percentages of 98.92%, 95.86%, and 87.30% for PhaP4Abs7, PhaP5Abs7, and the σ24 factor, respectively (Figure 2D–F). In the Ramachandran graphs, green regions represent the favored and allowed regions, which correspond to conformations that do not exist in steric shocks. The favored regions include the dihedral angles, typical of alpha-helix and beta-sheet conformations. White areas are related to conformations where atoms of the polypeptide come closer than the sum of their van der Waals ratios. Therefore, white areas are sterically forbidden regions for all amino acids except for glycine, which lacks a side chain [41].

Currently, the prediction, the search, and the recognition of the promoter region of the *pha*C gene to detect, *in silico*, the −10 and −35 boxes are challenging due to the diversity of genetic sequences housed in worldwide databases [18,21]. It is essential to know the binding sites of the sigma factors to the promoter region and the subsequent binding of the RNA polymerase to contribute to the beginning of DNA transcription. In *A. brasilense* Sp7, there is insufficient information about the σ^24^ factor and phasins. However, the continuous acquisition of servers and updating of the software will allow future research to be refined in order to obtain more detailed results.

This study shows that the interaction between phasins and the −10 and −35 boxes of the promoter region of the *pha*C gene is possible. The above was probed using the MEME Suite server. It is well known that recognition of the binding sites of sigma factors and RNA polymerase is critical for DNA transcription. Nevertheless, these motifs are poorly conserved for certain microorganisms, which implies that they differ even between the different sigma factors that recognize them [18,21,42]. There are well-characterized motifs for the −10 and −35 regions in *E. coli* for interaction with the σ^70^, σ^54^, σ^28^, and σ^32^ factors. Nonetheless, the motifs recognized by the σ^24^ and σ^28^ factors have not been characterized due to the low level of conservation between them. Considering the position of the canonical motifs at −10 and −35, our results showed two probable boxes detected upstream of the *pha*C gene: for the −10 box, the ACCGCAC sequence, and for the −35 box, the CCTAAA sequence. These box sequences are commonly found close to the gene transcription start site or in intergenic regions. The distance between the gene transcription start site and the −10 and −35 boxes can vary from 20 to 200 base pairs [42].

In this study, we analyzed the binding of the phasins PhaP4_Abs7_ and PhaP5_Abs7_ to the promoter region of the *pha*C gene, which codes for PHB synthase in *A. brasilense* Sp7. Phasins were involved in stabilizing the PHB inclusions in the cytoplasm due to their amphiphilic properties [4,5,43]. There has been insufficient research evaluating whether the interactions between phasins and DNA could promote or prevent DNA transcription or if their interaction with other components, such as the sigma factors, could affect DNA transcription. Most PHB-producing microorganisms synthesize phasins that bind to PHB. In several bacteria, such as *Ralstonia eutropha* and *Pseudomonas. putida, *phasins bind to DNA. The *R. eutropha* PhaM is involved in binding PHB granules to the nucleoid region. In *P. putida*, the C-terminal side of the PhaF phasin binds to DNA since it has a histone-like C-terminal domain with highly positively charged amino acid residues. The interaction between PhaF and DNA alters the transcription of the *pha*C gene because PhaF interacts with the major groove of the DNA and forms a super helix [6,44,45,46]. It has also been suggested that PhaF exhibits a structural disorder unless it is in the formation of a complex with DNA. The above could be due to the electrostatic repulsions generated by the highly charged C-terminal domain of PhaF [46]. Modulating of the actions of PHB synthesis in a PhaC-dependent manner by PhaP1_Reu_ has also been proposed in *R. eutropha* [6]. However, in *A. brasilense*, there is no information about how phasins might interact with other proteins or DNA.

The bioinformatics analyses of the molecular docking between phasins and DNA found a blind molecular docking and showed phasins bind through their C-terminal side to the −10 box, such as PhaP4_Abs7_. However, PhaP5_Abs7_ is secure in the −35 box of the *pha*C gene. This data could suggest a modification in the gene transcription process, like the *Pseudomonas* sp. PhaF, which controls the expression of the *pha*C1 and *pha*I genes [5]. In other microorganisms, such as *A. hydrophila*, it has been observed that the overexpression of the *pha*P gene increases the expression of the *pha*C gene [45]. In the case of the σ^24^ factor, it binds to the −35 box. Previous studies have shown that the σ^24^ factor binds to the −10 and −35 boxes of the *chr*R1 and *chr*R2 genes [17,19].

Likewise, the essential regions of the σ^24^ factor for binding and recognition of the promoter regions of the gene have been reported. Thus, region 2.3, which consists of the amino acids 68 to 87 of the σ^24^ factor, was involved in the correct interaction of the σ^24^ factor with the RNA polymerase. Region 2.4, involving the amino acids 88 to 101, recognizes the −10 box. Finally, region 4.2, formed by the amino acids 158 to 191, was involved in binding to the −35 box [17]. In the molecular docking of the phasin-σ^24^ factor complex, it was observed that PhaP4_Abs7_ does not bind to any region of interaction of the σ^24^ factor, whereas the PhaP5_Abs7_ phasin binds to these contact regions with two amino acids, one of them in the 2.4 region and the other one in the 4.2 region.

In the molecular docking of the protein complex with DNA, it was analyzed whether the σ^24^ factor still interacted with the reported regions in which it carries out recognition and subsequent binding to DNA. Consequently, it was observed that in the PhaP4_Abs7_-σ^24^ complex, there are no interactions with the regions, whereas in the PhaP5_Abs7_-σ^24^ complex, the region of the σ^24^ factor that interacts is the 2.1 region. In addition, PhaP5_Abs7_ had the highest number of attraction charges and hydrogen bonds, which could allow it to have better stabilization in the binding process in comparison with the phasin PhaP_4Abs7_.

Regarding the types of hydrogen bond interactions, we noticed that both servers identified the same amino acids involved in these interactions in molecular docking. We suggest that the servers are complementary to one another, based on the identification of similar amino acids. However, we also observed that there are ±2 different amino acids, which is a result of the algorithms that each server runs.

Evaluating the binding of phasins to this upstream region may be a key point in the transcription of the *pha*C gene and consequently in the production of the PHB. Since phasins do not bind to elemental regions in the σ^24^ factor, the results may indicate phasins could stabilize the RNA polymerase or even lead to correcting the interaction between the σ^24^ factor and DNA. The above could be explained by the phasins PhaP4_Abs7_ and PhaP5_Abs7_ in the protein complex with the σ^24^ factor not binding to the 2.4 or 4.2 regions of the σ^24^ factor. On the contrary, the phasins were involved in the detection of the −10 and −35 boxes.

Therefore, the increase in the transcription of the *pha*C gene would increase the PhaC protein, which is essential for PHB synthesis in *A. brasilense*. This could lead to the modification of the parameters of its industrial production and promote the change from single-use plastics (made from petroleum) to biodegradable plastics for use in the food industry and in health areas. In addition to being friendly to the environment, bioplastics could reduce the carbon footprint on the planet.

## 5. Conclusions

When the phasins PhaP4_Abs7_ and PhaP5_Abs7_ or the σ^24^ factor were individually coupled to DNA, they showed a stable interaction, granted by hydrogen bridges, salt bridges, and attractive charges, among others—interactions that could help stabilize the binding of proteins to DNA. The unique phasin that binds to a different region is the PhaP4_Abs7_ phasin, which interacts with the −10 box of the *pha*C gene. On the contrary, the PhaP5_Abs7_ phasin and the σ^24^ factor are bound close to the −35 box of the *pha*C gene. However, when the phasins form a protein complex with the σ^24^ factor (PhaP4_Abs7_-σ^24^ and PhaP5_Abs7_-σ^24^) and then the protein complex binds to DNA, the binding amino acids change as well as the nucleotides involved in their interaction. In the three evaluated situations, the protein complexes bind near the −35 box of the *pha*C gene. But with the PhaP4_Abs7_-σ^24^ protein complex, the phasin interacts with the two regions involved to detect the −10 and −35 boxes. Otherwise, the phasin does not interact with these regions.

For this reason, the binding of the phasins to promoter regions of the *pha*C gene, individually or by creating a protein complex with the σ^24^ factor, could be a key point in the transcription of the *pha*C gene and consequently affect the synthesis of PHB. It is necessary to carry out more analyses in the future to elucidate whether this effect can inhibit or promote the transcription of the *pha*C gene in *Azospirillum brasilense* Sp7.

## Figures and Tables

**Figure 1 polymers-16-00611-f001:**
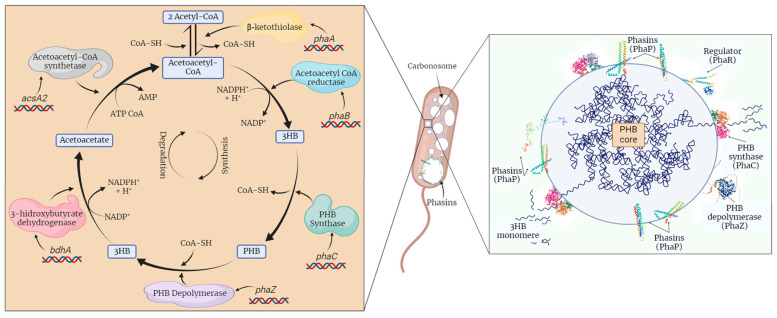
PHB biosynthesis pathway (**left**) and granule-associated proteins in carbonosome (**right**). PHB biosynthesis enzymes: acetyl-CoA acetyltransferase (PhaA) encoded by *pha*A gene, acetoacetyl-CoA reductase (PhaB) encoded by *pha*B gene, and PHA synthase (PhaC) encode by *pha*C gene. 3HB: 3 hydroxybutyryl; PHB: polyhydroxybutyrate; CoA-SH: Coenzyme A, NADPH; nicotinamide adenine dinucleotide phosphate; AMP: adenosine monophosphate (created on BioRender.com).

**Figure 2 polymers-16-00611-f002:**
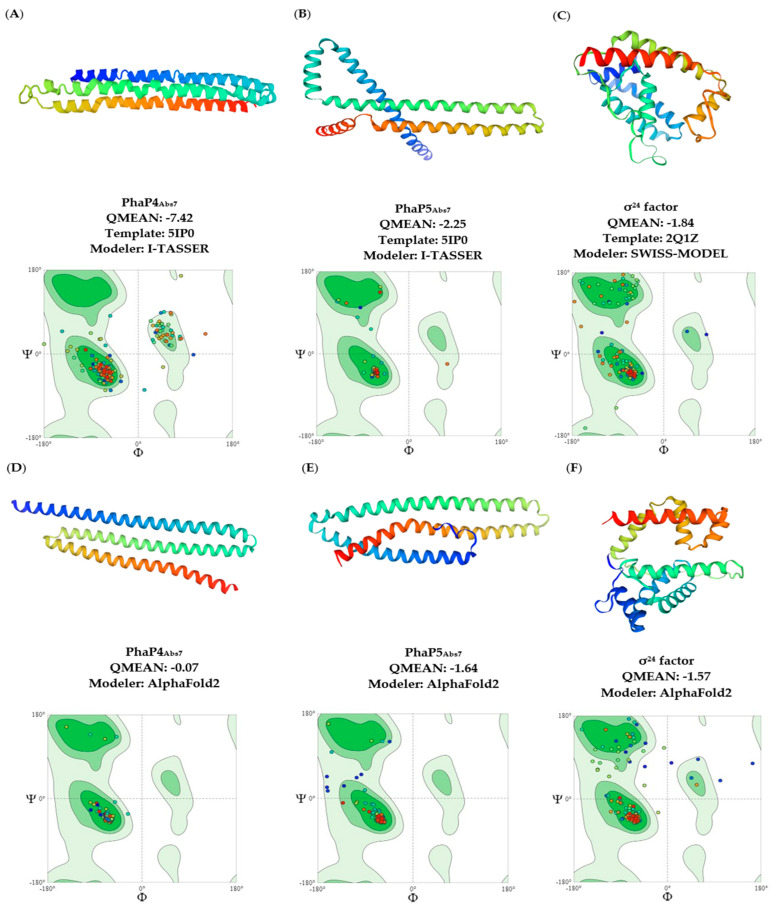
Prediction of three-dimensional structures of *A. brasilense* Sp7 phasins and σ^24^ factor. First, a three-dimensional prediction was made through the I-TASSER and SWISS-Model servers: (**A**) PhaP4_Abs7_, (**B**) PhaP5_Abs7_, (**C**) σ^24^ factor. Second, three-dimensional prediction was made through Alpha Fold2: (**D**) PhaP4_Abs7_, (**E**) PhaP5_Abs7_, (**F**) σ^24^ factor. The Ramachandran plots and the QMEAN values obtained are shown.

**Figure 3 polymers-16-00611-f003:**
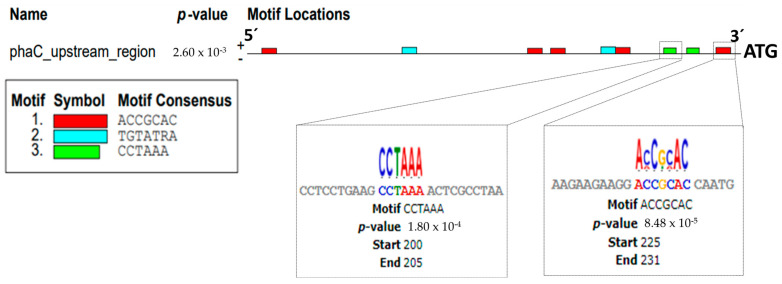
Motifs in the upstream region of the *pha*C gene, box −10 (green) and box −35 (red). The boxes were determined through the MEME suite server.

**Figure 4 polymers-16-00611-f004:**
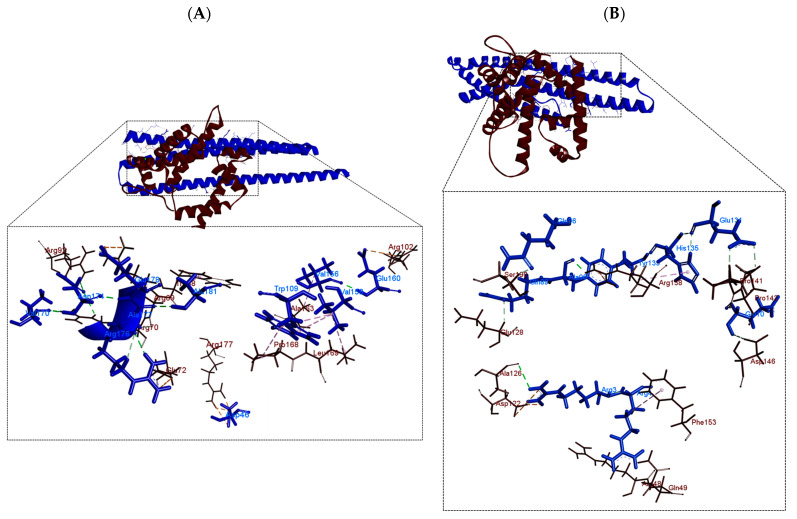
Molecular docking between phasin and σ^24^ factor. Models were obtained by ClusPro2.0; three-dimensional models and interactions were determined using the Biovia Discovery Visualizer. Phasins (blue) and σ^24^ factor (dark brown): (**A**) PhaP4_Abs7_-σ^24^ and (**B**) PhaP5_Abs7_-σ^24^.

**Figure 5 polymers-16-00611-f005:**
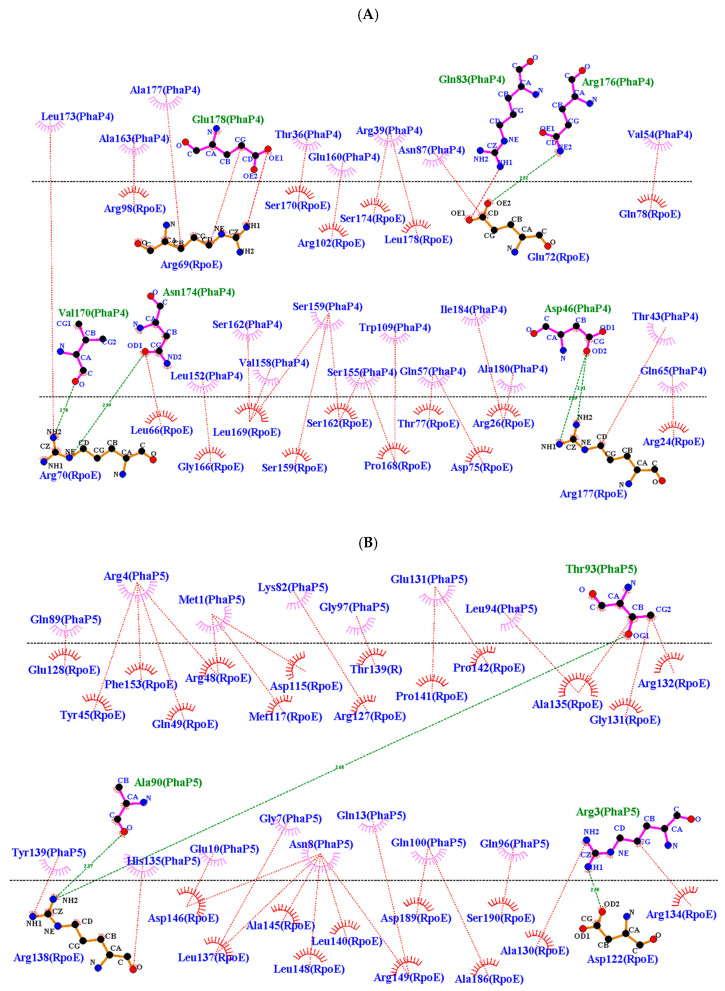
Two-dimensional diagrams of the phasin–sigma factor protein complexes. The diagrams were generated in LigPlot from molecular docking results obtained with ClusPro2.0. The hydrogen bonds formed between the amino acids of the phasin and the σ^24^ factor are represented with green dotted lines, and hydrophobic interactions are represented with red dotted lines. (**A**) PhaP4_Abs7_-σ^24^ and (**B**) PhaP5_Abs7_-σ^24^.

**Figure 6 polymers-16-00611-f006:**
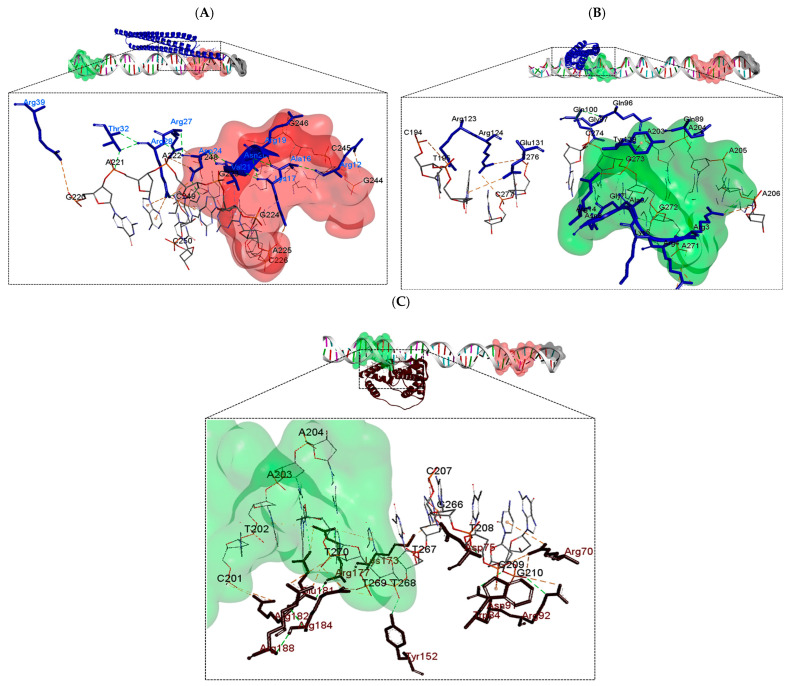
Molecular docking between phasin and DNA. Models were obtained by the HDOCK server, and three-dimensional models and interactions were determined using the Biovia Discovery Visualizer. Phasins (blue), σ^24^ factor (dark brown), −10 box (red), −35 box (green), and ATG (black): (**A**) PhaP4_Abs7_-DNA, (**B**) PhaP5_Abs7_-DNA, and (**C**) σ^24^-DNA.

**Figure 7 polymers-16-00611-f007:**
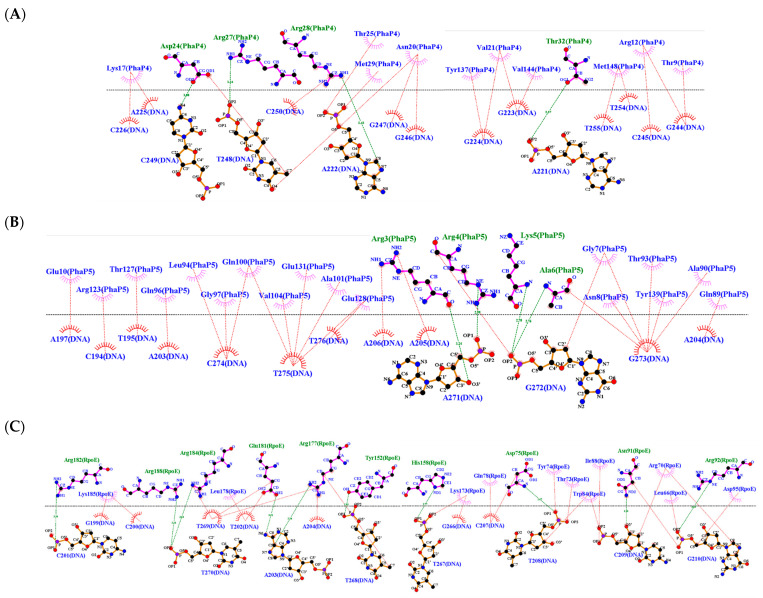
Two-dimensional diagrams of interactions between phasin and sigma factor with DNA. The diagrams were generated in LigPlot from the molecular docking results obtained with the HDOCK server. Hydrogen bonds formed between amino acids and nucleotides are represented with green dotted lines, and hydrophobic interactions are represented with red dotted lines. (**A**) PhaP4_Abs7_-DNA, (**B**) PhaP5_Abs7_- DNA, and (**C**) σ^24^-DNA.

**Figure 8 polymers-16-00611-f008:**
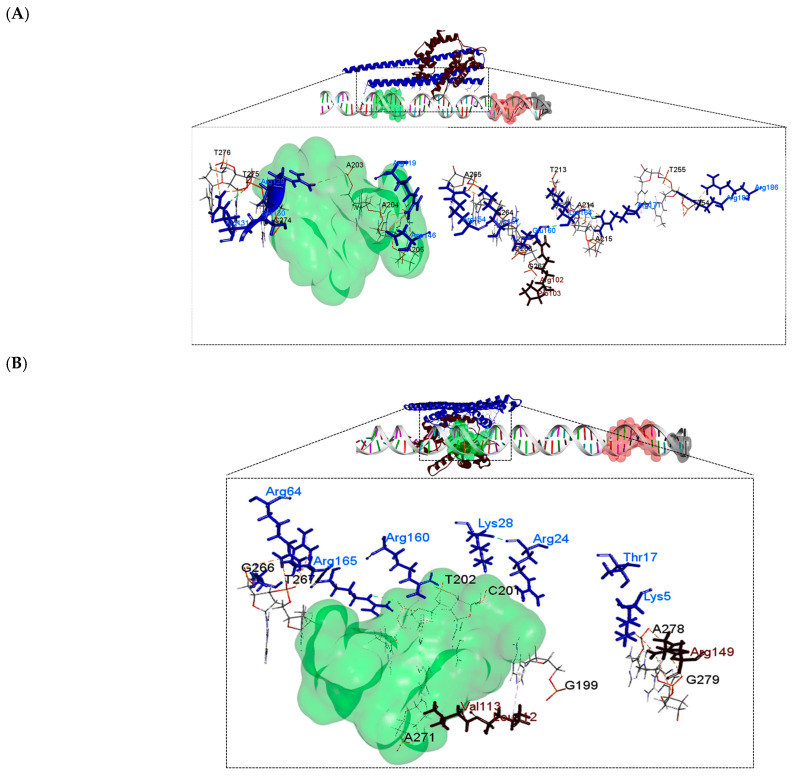
Molecular docking between the protein complex and DNA. Models were obtained by the HDOCK server; three-dimensional models and interactions were determined using the Biovia Discovery Visualizer. Phasins (blue), σ^24^ factor (dark brown), −10 box (red), −35 box (green) and ATG (black): (**A**) PhaP4_Abs7_-σ^24^-DNA and (**B**) PhaP5_Abs7-_σ^24^-DNA.

**Figure 9 polymers-16-00611-f009:**
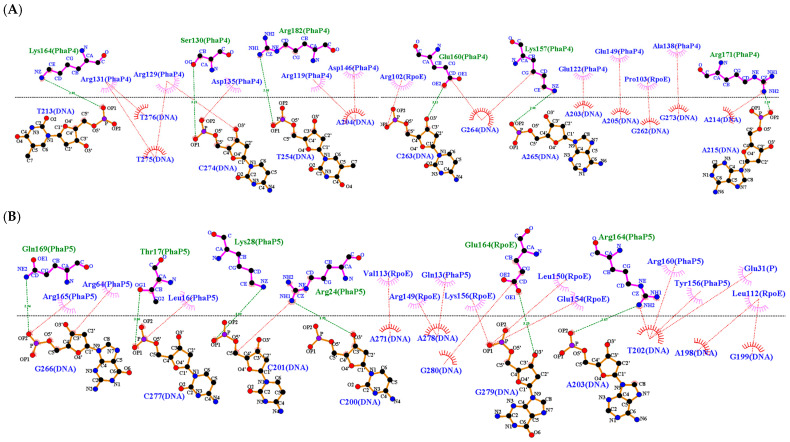
Two-dimensional diagrams of interactions between the protein complex and DNA. The diagrams were generated in LigPlot from the molecular docking results obtained with the HDOCK server. Hydrogen bonds formed between amino acids and nucleotides are observed in green dotted lines, and hydrophobic interactions are represented with red dotted lines. (**A**) PhaP4_Abs7_-σ^24^-DNA and (**B**) PhaP5_Abs7_-σ^24^-DNA.

**Table 1 polymers-16-00611-t001:** Hydrogen bonds detected from phasins in protein–protein molecular docking between phasin and σ^24^ factor. Hydrogen bonds were detected using the Biovia Discovery Visualizer and LigPlot.

Receptor	Ligand	Number of Hydrogen Bonds		Amino Acids with Hydrogen Interactions	
		Biovia	LigPlot	Biovia	LigPlot
Factor σ^24^	PhaP4_Abs7_	4	5	Gln83, Val170, Asn174	Asp46, Gln83, Val170, Asn174
	PhaP5_Abs7_	3	3	Arg3, Glu10, Ala90	Arg3, Ala90, Thr93

**Table 2 polymers-16-00611-t002:** Hydrogen bonds detected from phasins and σ^24^ factor with DNA in protein–DNA molecular docking. Hydrogen bonds were detected using the Biovia Discovery Visualizer and LigPlot.

Receptor	Ligand	Number of Hydrogen Bonds		Amino Acids with Hydrogen Interactions	
		Biovia	LigPlot	Biovia	LigPlot
DNA (upstream region of *pha*C)	PhaP4_Abs7_	3	4	Asp24, Arg28, Thr32	Asp24, Arg27, Arg28, Thr32
	PhaP5_Abs7_	5	4	Arg3, Arg4, Lys5, Ala6, Asn8	Arg3, Arg4, Lys5, Ala6
	Factor σ^24^	6	10	Asp75, Asn91, Arg92, Tyr152, Arg177, Glu181	Asp75, Asn91, Arg92, Tyr152, His158, Arg177, Glu181 Arg182, Arg184, Arg188

**Table 3 polymers-16-00611-t003:** Hydrogen bonds were detected between the protein complex (phasin-sigma factor) and DNA. Hydrogen bonds were detected using the Biovia Discovery Visualizer and LigPlot. * Phasin, ** σ^24^ factor.

Receptor	Ligand	Number of Hydrogen Bonds		Amino Acids with Hydrogen Interactions	
		Biovia	LigPlot	Biovia	LigPlot
DNA (upstream region of *pha*C)	PhaP4_Abs7_-σ^24^	3	6	Ser130 *, Arg131 *, Arg171 *	Ser130 *, Lys157 *, Glu160 *, Lys164 *, Arg171 *, Arg182 *
	PhaP5_Abs7_-σ^24^	3	6	Thr17 *, Arg164 *, Gln169 *	Thr17 *, Arg24 *, Lys28 *, Gln169 *, Arg164 *, Glu164 **

## Data Availability

Data are contained within the article.

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
