# Peer review of "Computational Analysis of the Tripartite Interaction of Phasins (PhaP4 and 5)-Sigma Factor (σ24)-DNA of Azospirillum brasilense Sp7"

_polymers, 2024, doi:10.3390/polym16050611_

Round 1

Reviewer 1 Report

Comments and Suggestions for Authors

In the absence of experimental data, only molecular modeling methods can provide initial information of the structure of macromolecular complexes. The authors made the first attempt to obtain the structures of A. brasilense phasins and their possible complexes with the promoter region of the phaC gene and with the σ24 factor with the help in silico tools. Of course, such kind of work is of a big interest from a fundamental and practical point of view. The main weakness of the paper is the lack of validation of the resulting models or at least some attempt to compare the results with known experimental data.

Below is a list of comments and questions for the authors.

Introduction

- The introduction is very long and contains a lot of details that are only indirectly related to the matter. For example, lines 99-108 are a detailed description of the mechanism of interaction between a phasin and DNA of another bacterium; such detailed description would be more appropriate in the Discussion when perform a comparative analysis of the obtained results with the literature data. Similarly, lines 84-92 provide a detailed description of the bacteria that are not the object of the study. The introduction needs to be shortened, irrelevant information should be transferred to the discussion or omitted.

- Figure 1 should be improved. It says nothing about the objects of the study (phasins, sigma factor and DNA), and at the same time, all the details of the PHB synthesis cycle are given. Perhaps these details are appropriate for “Polymers,” however the figure needs to be supplemented by showing the role of phasins and the sigma factor in the cycle of PHB synthesis. Depiction of abbreviations should be added to the capture of Figure 1.

Methods

- PDB structures 5IP0 and 2Q1Z should be cited (see citation policies https://www.rcsb.org/pages/policies#citationPolicies)

- Which chains of 5IP0 and 2Q1Z were used as the templates? Chains A and C of 5IP0 have different conformations.

Reuslts

- The considerable difference between the structure of PhaP6Abs7 and the structures of PhaP4Abs7, PhaP5Abs7 and 5IP0 is striking. Such a large difference, coupled with the poor score of the model, raises doubts about its reliability and must be explained. How different are the primary sequences of PhaP4Abs7, PhaP5Abs7, PhaP6Abs7, and 5IP0?

- A comparative conformational analysis of the resulting PhaP4Abs7, PhaP5Abs7, PhaP6Abs7 abd sigma24 models has not been performed. Thus, the authors of structure 5IP0 (10.1038/SREP39424) showed that the phasin monomer PhaP from Aeromonas hydrophila can exist in two conformations; depending on the conformation, different amino acids take part in intramolecular interactions. What intramolecular interactions are there in helical structures PhaP4Abs7 and PhaP5Abs7? How similar are they to the interaction in 5IP0? And what about the disordered PhaP6Abs7 model ? Are there any intramolecular interactions in the model that can somehow keep it in the conformation found by the authors?

- It is surprising that the authors, having received an unsatisfactory score of the PhaP4Abs7 and PhaP6Abs7 models, continued to work with them (especially with the suspicious PhaP6Abs7 model). The authors need to try other software (may be AlphaFold, IntFOLD, HH-suite, RaptorX). Or, using homologous modeling, build models PhaP4Abs7 and PhaP6Abs7 based on the “successful” model PhaP5Abs7. Or optimize these models using molecular dynamics simulation. When a satisfactory QMEAN score is achieved, the docking procedure can be performed.

- If the structure of PhaP6Abs7 is really so disordered (which I still doubt; rather, the model is built incorrectly), then rigid docking of this model is completely pointless. In solution, this structure will have a countless number of conformations and the reliability of the DNA-PhaP6Abs7, sigma-PhaP6Abs7 and DNA-sigma-PhaP6Abs7 complexes obtained by the authors is zero. If the authors fail to improve the PhaP6Abs7 model, then they will have to abandon this object and limit the study to only PhaP4Abs7 and PhaP5Abs7. Or the authors must conduct a long MD simulation (minimum 200 ns) of the conformational behavior of PhaP6Abs7 in solution and prove that the found conformation is stable.

- Figure 3 shows the TGTATRA motif, which is not described anywhere in the text.

- Lines 208-209. It can not be stated that the interaction is stable unless this stability is verified by the molecular dynamic simulation.

- Line 211-212. The supplementary material is not provided.

- Lines 410-413. PhaP5Abs7 is mentioned twice. In comparison with phasins PhaP4Abs7 and PhaP6Abs7?

Author Response

Dear reviewers, I appreciate the time and dedication to the manuscript Computational analysis of the tripartite interaction of phasins (PhaP4 and 5)-Sigma factor (σ24)-DNA of Azospirillum brasilense Sp7. All your questions have been answered, and concerns are marked in yellow for easy location in the body of the manuscript. I complied with everything requested and hope the manuscript is accepted for publication

Reviewer 1

Comments and Suggestions for Authors

In the absence of experimental data, only molecular modeling methods can provide initial information of the structure of macromolecular complexes. The authors made the first attempt to obtain the structures of A. brasilense phasins and their possible complexes with the promoter region of the phaC gene and with the σ24 factor with the help in silico tools. Of course, such kind of work is of a big interest from a fundamental and practical point of view. The main weakness of the paper is the lack of validation of the resulting models or at least some attempt to compare the results with known experimental data.

We did not find reported data on these complexes in Azospirillum brasilense, our laboratory has been working on this topic since 2019 and we are carrying out trials that are not yet mature.

Below is a list of comments and questions for the authors.

Introduction

- The introduction is very long and contains a lot of details that are only indirectly related to the matter. For example, lines 99-108 are a detailed description of the mechanism of interaction between a phasin and DNA of another bacterium; such detailed description would be more appropriate in the Discussion when perform a comparative analysis of the obtained results with the literature data. Similarly, lines 84-92 provide a detailed description of the bacteria that are not the object of the study. The introduction needs to be shortened, irrelevant information should be transferred to the discussion or omitted.

The introduction was shortened and modified giving more clarity to the writing

-Figure 1 should be improved. It says nothing about the objects of the study (phasins, sigma factor and DNA), and at the same time, all the details of the PHB synthesis cycle are given. Perhaps these details are appropriate for “Polymers,” however the figure needs to be supplemented by showing the role of phasins and the sigma factor in the cycle of PHB synthesis. Depiction of abbreviations should be added to the capture of Figure 1.

Was done

Figure 1. PHB biosynthesis pathway (left) and granule-associated proteins in carbonosome (right). PHB biosynthesis enzymes: acetyl-CoA acetyltransferase (PhaA) encoded by phaA gene, acetoacetyl-CoA reductase (PhaB) encoded by phaB gene and PHA synthase (PhaC) encode by phaC gene. 3HB: 3 hydroxybutyryl, PHB: polyhydroxybutyrate, CoA-SH: Coenzyme A, NADPH: Nicotinamide adenine dinucleotide phosphate, AMP: Adenosine monophosphate, (Created on BioRender.com].

Methods

- PDB structures 5IP0 and 2Q1Z should be cited (see citation policies https://www.rcsb.org/pages/policies#citationPolicies)

Was done.

The three-dimensional structures of A. brasilense Sp7, PhaP4Abs7, and PhaP5Abs7 were predicted by the Iterative Threading ASSEmbly Refinement (I-TASSER) server [22–25]. The crystallized structure of phasin PhaPAhy of Aeromonas hydrophila, chain A (PDB number-5IP0) was used as a template [26]. For the σ24 factor, the three-dimensional structure was generated using the SWISS-MODEL server [27]. The crystallized structure of Rhodobacter sphaeroides SigE, chain A (PDB number-2Q1Z) was used as a template [28]. As second step using Alpha Fold2 server three-dimensional structures of the three protein were generated [29], to compare model between servers and select the models with the best values to proceed with the molecular dockings. Using the SWISS-MODEL structure evaluation server, a Qualitative Model Energy ANalysis (QMEAN) and its Ramachandran plots were performed to evaluate the quality of the three-dimensional structures [30]

- Which chains of 5IP0 and 2Q1Z were used as the templates? Chains A and C of 5IP0 have different conformations.

Was done

The three-dimensional structures of A. brasilense Sp7, PhaP4Abs7, and PhaP5Abs7 were predicted by the Iterative Threading ASSEmbly Refinement (I-TASSER) server [22–25]. The crystallized structure of phasin PhaPAhy of Aeromonas hydrophila, chain A (PDB number-5IP0) was used as a template [26]. For the σ24 factor, the three-dimensional structure was generated using the SWISS-MODEL server [27]. The crystallized structure of Rhodobacter sphaeroides SigE, chain A (PDB number-2Q1Z) was used as a template [28]. As second step using Alpha Fold2 server three-dimensional structures of the three protein were generated [29], to compare model between servers and select the models with the best values to proceed with the molecular dockings. Using the SWISS-MODEL structure evaluation server, a Qualitative Model Energy ANalysis (QMEAN) and its Ramachandran plots were performed to evaluate the quality of the three-dimensional structures [30]

Reuslts

- The considerable difference between the structure of PhaP6Abs7 and the structures of PhaP4Abs7, PhaP5Abs7 and 5IP0 is striking. Such a large difference, coupled with the poor score of the model, raises doubts about its reliability and must be explained. How different are the primary sequences of PhaP4Abs7, PhaP5Abs7, PhaP6Abs7, and 5IP0?

At reviewer's suggestion, PhaP6 was removed.

- A comparative conformational analysis of the resulting PhaP4Abs7, PhaP5Abs7, PhaP6Abs7 abd sigma24 models has not been performed. Thus, the authors of structure 5IP0 (10.1038/SREP39424) showed that the phasin monomer PhaP from Aeromonas hydrophila can exist in two conformations; depending on the conformation, different amino acids take part in intramolecular interactions. What intramolecular interactions are there in helical structures PhaP4Abs7 and PhaP5Abs7? How similar are they to the interaction in 5IP0? And what about the disordered PhaP6Abs7 model ? Are there any intramolecular interactions in the model that can somehow keep it in the conformation found by the authors?

Was done.

The QMEAN results were acceptable for PhaP5Abs7 and the σ24 factor with a value of -2.25 and -1.84, respectively. However, for PhaP4Abs7 the quality was not as good as was expected, but acceptable (Figure 2 A-C). Nevertheless, three-dimensional models performed with Alpha Fold2 obtained better QMEAN scores, in comparison with models from I-TASSER and SWISS-MODEL, with values of -0.07 for PhaP4Abs7, -1.64 for PhaP5Abs7, and -1.57 for σ24 factor (Figure 2 D-F). For this reason, for protein-protein and protein-DNA molecular docking was performed using three-dimensional models getting by Alpha Fold2. Finally, a structural superimposition was performed, and suggesting that the two three-dimensional models of each protein obtained by the servers might be structurally very similar (supplementary material 1).

Simultaneously, a multiple alignment was carried out in Clustal omega between the A. brasilense Sp7 phasins and Aeromonas hydrophila phasin, (PDB number 5IP0). Results showed that A. brasilense phasins have a percentage identity of 11.90% for PhaP4Abs7 and 25.49% for PhaP5Abs7, respectively, to the Aeromonas hydrophila phasin. PhaP4Abs7, PhaP5Abs7, and PhaPAhy are 188, 171, and 116 amino acids in length, respectively. In addition, A. brasilense σ24 factor shows a 43.48% of identity to Rhodobacter sphaeroides SigE (supplementary material 1).

- It is surprising that the authors, having received an unsatisfactory score of the PhaP4Abs7 and PhaP6Abs7 models, continued to work with them (especially with the suspicious PhaP6Abs7 model). The authors need to try other software (may be AlphaFold, IntFOLD, HH-suite, RaptorX). Or, using homologous modeling, build models PhaP4Abs7 and PhaP6Abs7 based on the “successful” model PhaP5Abs7. Or optimize these models using molecular dynamics simulation. When a satisfactory QMEAN score is achieved, the docking procedure can be performed.

The analysis was carried out with Alpha Fold2 and also the PhaP6 protein came out with unsatisfactory results and was eliminated.

Figure 2. Prediction of three-dimensional structures of A. brasilense phasins Sp7 and σ24 factor. First three-dimensional prediction was made through the I-TASSER and SWISS-Model servers, A) PhaP4Abs7, B) PhaP5Abs7, C) σ24 factor. Second three-dimensional prediction was made through Alpha Fold2; D) PhaP4Abs7, E) PhaP5Abs7, F) σ24 factor the Ramachandran plots and the QMEAN values obtained are shown.

- If the structure of PhaP6Abs7 is really so disordered (which I still doubt; rather, the model is built incorrectly), then rigid docking of this model is completely pointless. In solution, this structure will have a countless number of conformations and the reliability of the DNA-PhaP6Abs7, sigma-PhaP6Abs7 and DNA-sigma-PhaP6Abs7 complexes obtained by the authors is zero. If the authors fail to improve the PhaP6Abs7 model, then they will have to abandon this object and limit the study to only PhaP4Abs7 and PhaP5Abs7. Or the authors must conduct a long MD simulation (minimum 200 ns) of the conformational behavior of PhaP6Abs7 in solution and prove that the found conformation is stable.

Thanks for your advice, PhaP6 has been removed.

- Figure 3 shows the TGTATRA motif, which is not described anywhere in the text.

It appears as a result of the analysis but was not used.

- Lines 208-209. It can not be stated that the interaction is stable unless this stability is verified by the molecular dynamic simulation.

Was done.

- Line 211-212. The supplementary material is not provided.

There must be an error, if it was provided, it is now two supplementary material files.

- Lines 410-413. PhaP5Abs7 is mentioned twice. In comparison with phasins PhaP4Abs7 and PhaP6Abs7?

It was fixed.

Reviewer 2 Report

Comments and Suggestions for Authors

The paper addresses a crucial topic, but the modeling exercise appears to have low quality. It raises questions about the choice of ITASSER and Swiss-model over the state-of-the-art Alphafold tool for structure prediction. It is suggested to model using Alphafold and compare the new results with the existing generated models.

The method employed for docking experiments is unclear. There is no indication of whether the authors used blind docking or information-driven docking. The text lacks an explanation in this regard.

Tables present Hydrogen bonds derived from two different software tools, but the differences in the number of H-bonds are not justified. It is crucial to provide explanations for these variations.

The overall quality of the figures is poor and needs improvement. It is recommended to redraw all figures for clarity.

Lastly, having a native English speaker review the manuscript is advisable, as the current format lacks clarity.

Comments on the Quality of English Language

Very poor. 

Author Response

Dear reviewers, I appreciate the time and dedication to the manuscript Computational analysis of the tripartite interaction of phasins (PhaP4 and 5)-Sigma factor (σ24)-DNA of Azospirillum brasilense Sp7. All your questions have been answered, and concerns are marked in yellow for easy location in the body of the manuscript. I complied with everything requested and hope the manuscript is accepted for publication

Reviewer 2

Comments and Suggestions for Authors

The paper addresses a crucial topic, but the modeling exercise appears to have low quality. It raises questions about the choice of ITASSER and Swiss-model over the state-of-the-art Alphafold tool for structure prediction. It is suggested to model using Alphafold and compare the new results with the existing generated models.

Alpha Fold2 was also used.

The method employed for docking experiments is unclear. There is no indication of whether the authors used blind docking or information-driven docking. The text lacks an explanation in this regard.

Existing information from two different bacteria was used.

Simultaneously, a multiple alignment was carried out in Clustal omega between the A. brasilense Sp7 phasins and Aeromonas hydrophila phasin, (PDB number 5IP0). Results showed that A. brasilense phasins have a percentage identity of 11.90% for PhaP4Abs7 and 25.49% for PhaP5Abs7, respectively, to the Aeromonas hydrophila phasin. PhaP4Abs7, PhaP5Abs7, and PhaPAhy are 188, 171, and 116 amino acids in length, respectively. In addition, A. brasilense σ24 factor shows a 43.48% of identity to Rhodobacter sphaeroides SigE (supplementary material 1).

Tables present Hydrogen bonds derived from two different software tools, but the differences in the number of H-bonds are not justified. It is crucial to provide explanations for these variations.

Was done.

The overall quality of the figures is poor and needs improvement. It is recommended to redraw all figures for clarity.

Was done.

Lastly, having a native English speaker review the manuscript is advisable, as the current format lacks clarity.

Comments on the Quality of English Language

Very poor.

English was revised.

Round 2

Reviewer 1 Report

Comments and Suggestions for Authors

The authors have revised the article properly, it can be published in present form.

Author Response

Thanks.

Reviewer 2 Report

Comments and Suggestions for Authors

Please redraw all figures. They are not clear at all. 

Comments on the Quality of English Language

It can be improved

Author Response

Reviewer 2

All figures were made again, paying attention to the size of numbers and letters, as well as the resolution.

English was improved.

Round 3

Reviewer 2 Report

Comments and Suggestions for Authors

Acceptable

Comments on the Quality of English Language

It is ok now.